# Identification of a novel path for cerebrospinal fluid (CSF) drainage of the human brain

Joel E. Pessa 🅰 *

Private Practice, Arlington, Massachusetts, United States of America

* jepessa@gmail.com

**Data Availability Statement:** The DATA is included within the paper.

**Funding:** The author(s) received no specific funding for this work.

## Abstract

How cerebrospinal fluid (CSF) drains from the human brain is of paramount importance to cerebral health and physiology. Obstructed CSF drainage results in increased intra-cranial pressure and a predictable cascade of events including dilated cerebral ventricles and ultimately cell death. The current and accepted model of CSF drainage in humans suggests CSF drains from the subarachnoid space into the sagittal sinus vein. Here we identify a new structure in the sagittal sinus of the human brain by anatomic cadaver dissection. *The CSF canalicular system* is a series of channels on either side of the sagittal sinus vein that communicate with subarachnoid cerebrospinal fluid via Virchow-Robin spaces. Fluorescent injection confirms that these channels are patent and that flow occurs independent of the venous system. Fluoroscopy identified flow from the sagittal sinus to the cranial base. We verify our previous identification of CSF channels in the neck that travel from the cranial base to the subclavian vein. Together, this information suggests a novel path for CSF drainage of the human brain that may represent the primary route for CSF recirculation. These findings have implications for basic anatomy, surgery, and neuroscience, and highlight the continued importance of gross anatomy to medical research and discovery.

## Introduction

How cerebrospinal fluid (CSF) drains from the human brain is of paramount importance to cerebral health and physiology. Obstructed CSF drainage results in increased intra-cranial pressure and a predictable cascade of events including dilated cerebral ventricles (hydrocephalus) and cell death [1, 2]. This paper identifies a new structure in the sagittal sinus of the human brain for CSF drainage, and confirms our previous findings of a CSF system in the neck. Together, this work suggests a novel path for CSF drainage of the human brain that could represent the primary path for CSF recirculation.

CSF is an ultra-filtrate of blood that has several functions [3–6]. CSF produced in the ventricles serves as a reservoir across which the transport of energy molecules and solute occurs [3–6]. CSF accumulates in the subarachnoid space where it acts as a shock absorber protecting the brain from impact.

**Competing interests:** The authors have declared that no competing interests exist.

The current and accepted model of CSF drainage in humans suggests CSF drains from the subarachnoid space into the sagittal sinus vein (Figs 1 and 2) [7–10]. CSF is thought to diffuse through protrusions of the arachnoid meninges called "arachnoid granulations" or "villi" [11]. Although the number and distribution of arachnoid granulations is highly variable, this is the accepted route for CSF drainage in humans [12]. CSF also drains to lymphatic vessels in the nose (via the cribiform plate) most likely as minor pathway. Researchers identified an alternative route for CSF drainage in mice that involves CSF flow from dural lymphatics to the the scalp and neck [13, 14].

Our previous work identified the CSF drainage of the neck [15]. This present study identifies CSF channels in the sagittal sinus of the brain and completes the circuit for CSF drainage from the subarachnoid space to the subclavian vein.

## Materials and methods

Fresh cadaver dissections (not embalmed) were performed under institutional guidelines. The IRB committee reviewed this work and agreed that it does not require IRB approval or oversight. Seven (N = 7) dissections were performed in 5 male and 2 female cadavers. Ages ranged from 40–92 years (mean 82.5). The sagittal sinus was dissected to identify the bilateral CSF canicular system, and photographs were obtained in selected specimens (Canon EOS 6D 50mm macro lens, Canon USA, Melville, NY). Fluoroscopy was performed after the CSF canicular system was intubated with 0.25 mm silastic tubing (WPI, Sarasota, Florida) and injected with 0.2 mls of Omnipaque™ (iohexol, GE Healthcare, Chicago, Illinois). The CSF channels in the sagittal sinus were injected in 1 specimen using 1–5μ fluorescent polymer spheres (Cospheric™, Santa Barbara, CA) after intubation with 22/24 gauge IV catheter (Baxter

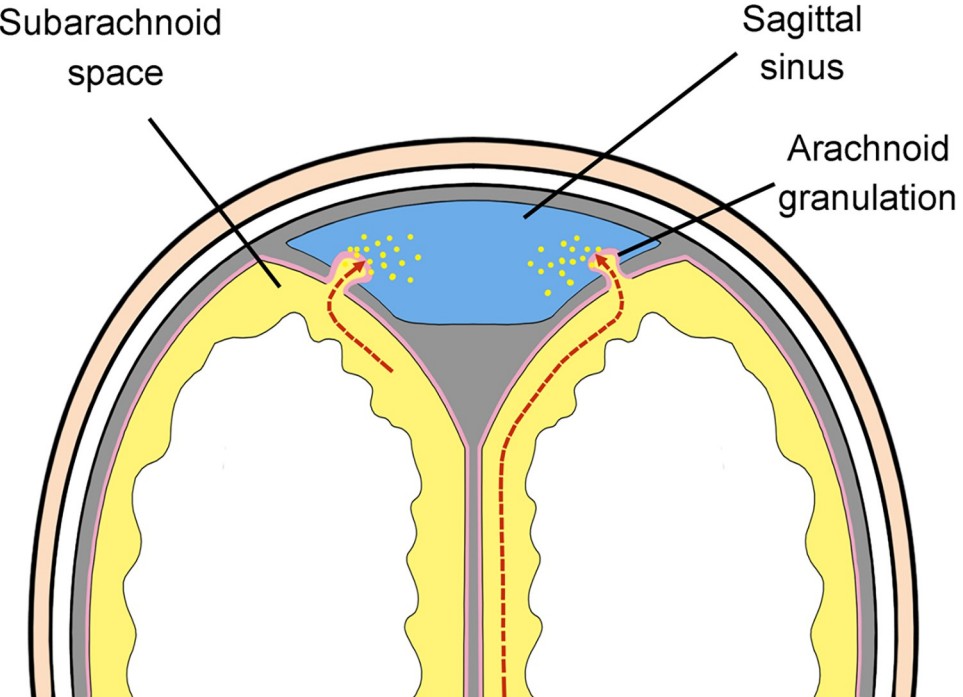

**Fig 1. Cross-section diagram of the accepted model of CSF drainage in humans.** CSF (yellow) in the subarachnoid space is thought to diffuse through protrusions of the arachnoid meninges (arachnoid granulations) into the sagittal sinus vein (blue).

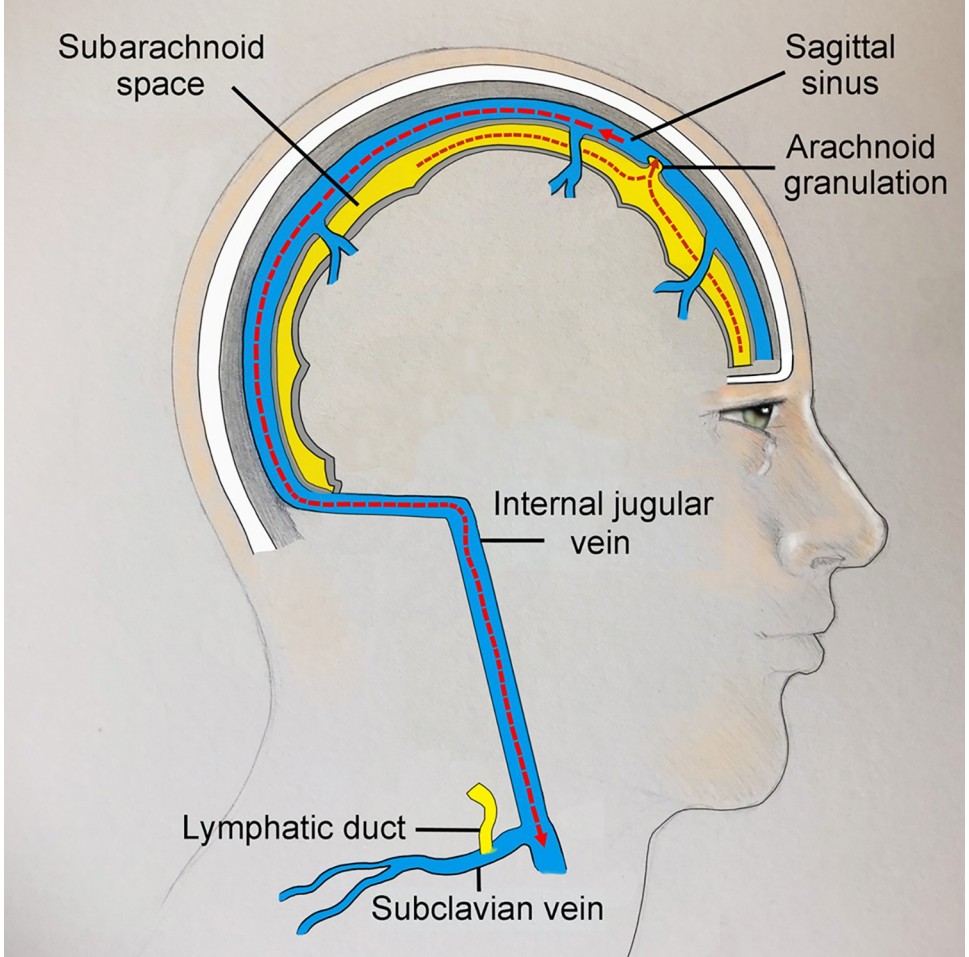

**Fig 2. Sagittal view of the current model of CSF drainage.** CSF (yellow) drains from the subarachnoid space through arachnoid granulations into the sagittal sinus vein (blue). Sagittal sinus venous blood exits the cranium through the jugular foramen to enter the internal jugular vein.

Healthcare Corporation, Deerfield, Illinois) and photographed under ambient/UV light. Neck dissections confirmed the anatomy of the CSF system in the neck in each specimen. One cervical biopsy was submitted for immunohistochemistry by 2-step indirect immunohistochemistry (IHC) for LYVE-1, D2-40, CD105, and F-actin (Sigma Aldrich, St. Louis, Missouri), and by direct IHC for the type-3 neurofilament protein vimentin (vimentin cy-3; Sigma Aldrich, St. Louis, Missouri) [15].

## Results

Craniotomy exposed the dura and the midline venous sagittal sinus (Fig 3). After the sagittal sinus was opened, blood was evacuated to identify CSF channels located on either side of the venous sagittal sinus (Figs 4 and 5). A cross-section view shows the location of CSF channels relative to the venous sinus (Fig 6). CSF channels exist as a plexus that are located posteriorly and on either side of the venous system (Fig 7). Dissection identified the CSF system in all specimens.

Fluoroscopy was performed in one specimen after the CSF channels on one side were injected with contrast material (Fig 8). Fluoroscopy confirmed flow in the CSF system (Fig 9).

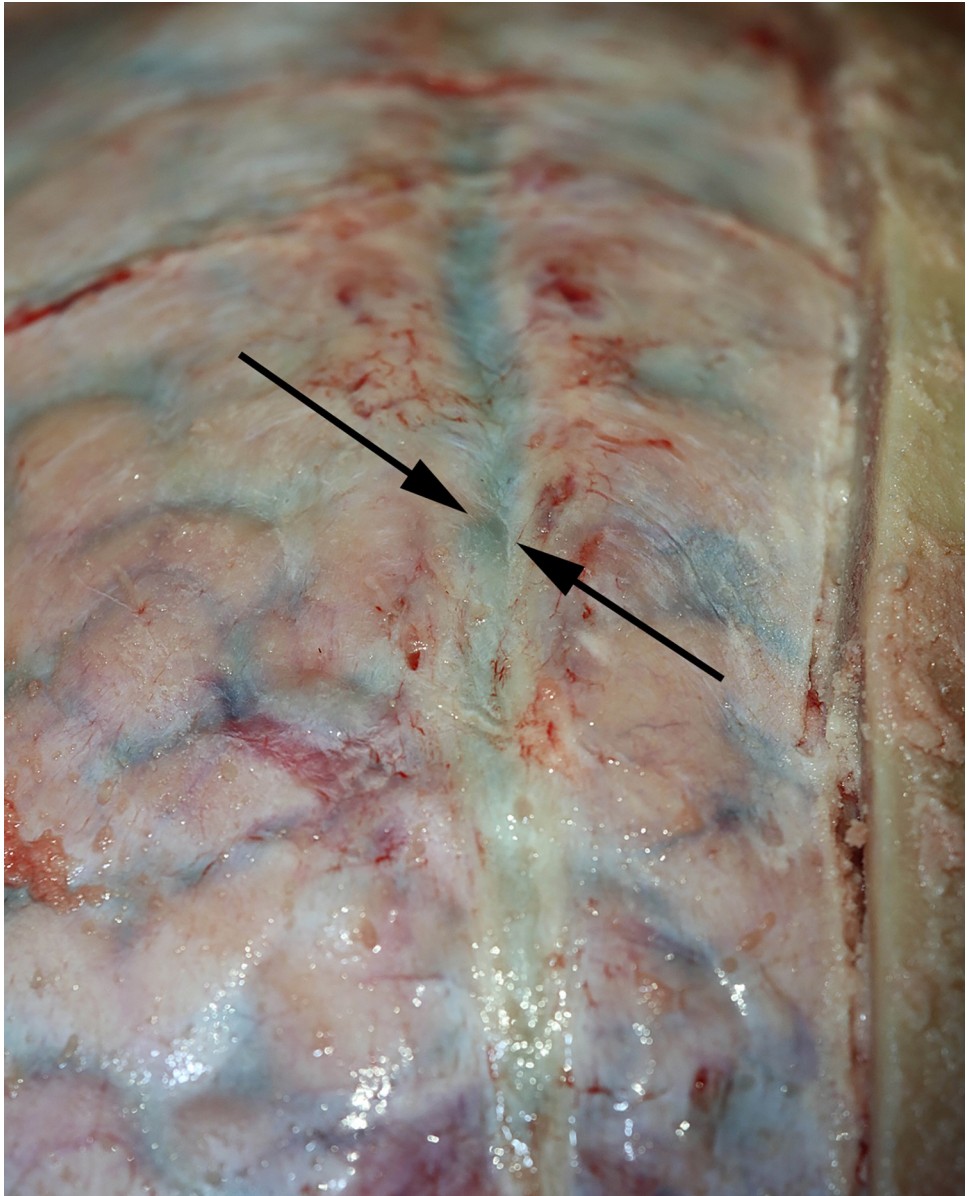

**Fig 3. Demonstration of the venous sagittal sinus.** The sagittal sinus is a midline venous structure (between arrows) in the dura. CSF channels travel on either side of this venous sinus.

Real-time imaging showed dye traveling from the superior sagittal sinus along the cranial base to exit the skull through the temporal fossa (Fig 9). There was no extravasation of dye (Fig 9), although CSF channels in the arachnoid meninges appeared to fill in a retrograde fashion (Fig 9).

We used a fluorescent dye to confirm flow in one specimen. The CSF channels were identified and cannulated with intravenous catheters (Fig 10) and Cospheric™ dye was injected to show flow (Fig 11). Imaging with UV light confirmed dye flow in the CSF system (Fig 12). There appeared to be some back flow into the arachnoid meninges (Fig 12). Dye injected into CSF channels did not fill the venous sagittal sinus or peripheral veins (Fig 13).

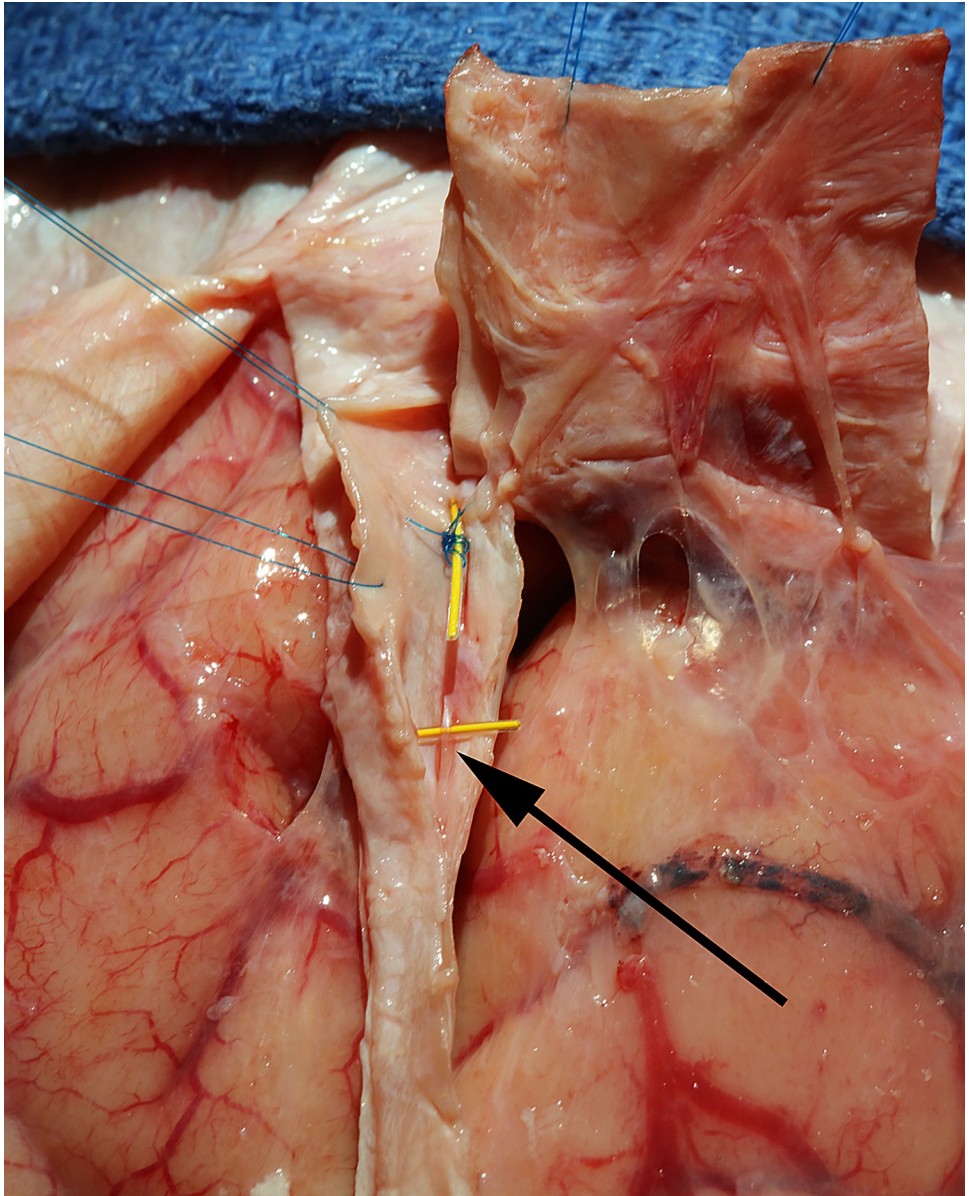

**Fig 4. CSF channels in the sagittal sinus.** Blood was evacuated from the venous sagittal sinus and is held open with blue sutures. The right CSF channel system (arrow) is identified over the yellow marker in a 90's year-old female specimen.

Immunohistochemistry was performed on a section of CSF channels harvested from the distal neck. Sections were negative for lymphatic markers LYVE-1 (Fig 14), and negative for vascular endothelial marker CD105 (Fig 15). This specimen was positive for vimentin, and suggested that CSF channels travel as a plexus (Fig 16).

Anatomical dissection verified the previous finding of CSF channels in the neck. The cervical CSF system travels in the carotid sheath (Fig 17). CSF channels are located in the outer layer (adventia) of the posterior internal jugular vein (Fig 17). Another dissection identifies CSF channels traveling in the carotid sheath (Fig 18).

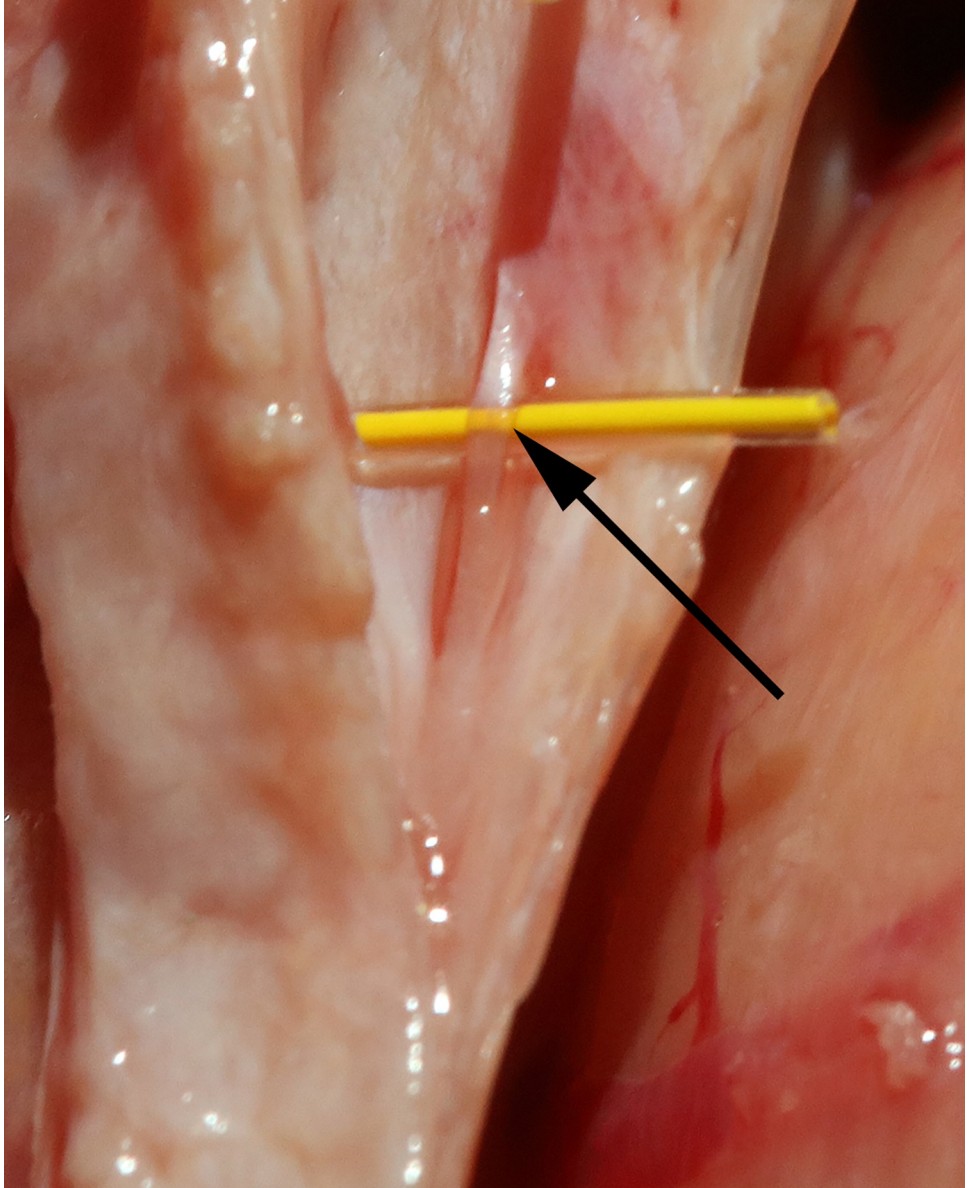

**Fig 5. CSF channels in the sagittal sinus.** This is a macro view of CSF channels (arrow) in the sagittal sinus.

## Discussion

This paper identifies a novel anatomical path for CSF drainage of the human brain. CSF channels in the superior sagittal sinus have not been previously described [7–12, 16–19]. Because CSF channels travel as a group or plexus and are embedded in surrounding tissue, this system is named *the CSF canalicular system*. CSF drainage by this route is independent of the venous sagittal sinus (Figs 19 and 20). CSF channels in the sagittal sinus are a consistent anatomical finding (Figs 21 and 22),

The CSF canalicular system is characterized as follows. CSF channels travel as a group or plexus, and appear as thin, translucent vessels that histologically lack a muscular layer. CSF drainage is privileged and does not involve intermediary lymphatic vessels or veins. CSF drains

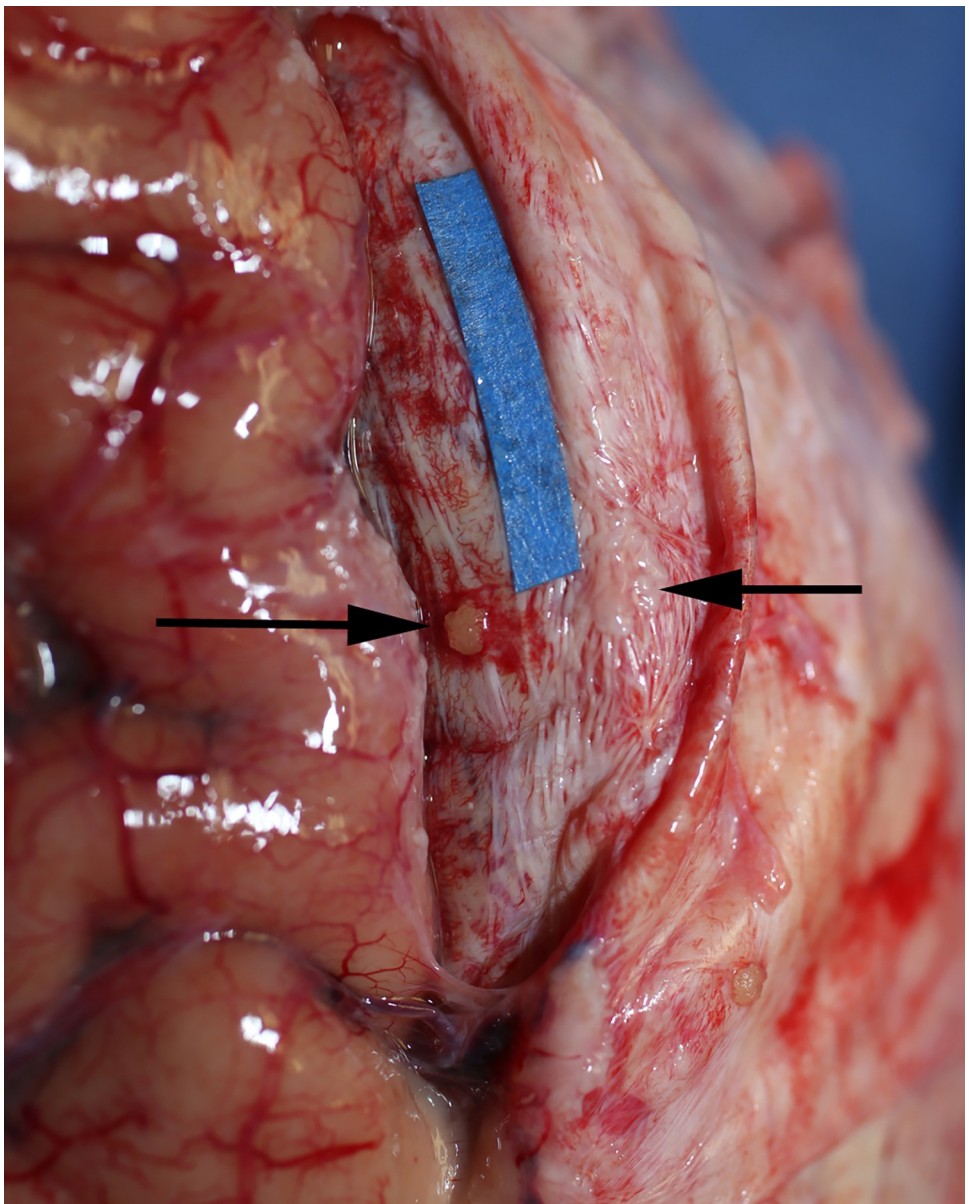

**Fig 6. Identification of the venous sagittal sinus.** The sagittal sinus lies within the dura (beneath blue marker) of the falx cerebri. Tumor (black arrows) has metastasized to the dura and arachnoid meninges. 80's year-old female specimen.

in a direct route from the arachnoid meninges to the subclavian vein, and has a similar terminal path to that described for peripheral nerves [15, 19]. It is noteworthy that CSF channels in the neck were probably described by Cruickshank and Mascagni in 1786 and 1787 respectively, although CSF vessels were mischaracterized as lymphatics [20, 21]. The features of the CSF canalicular system are summarized in Table 1.

The limitation of this work is the same as any post-mortem anatomical study, and is unable to evaluate function. With that caveat, the author suggests that the CSF canalicular system may be the primary path for CSF drainage in humans. Nerves are sensitive to pressure, and CSF drainage appears to be a redundant design in both the central and peripheral nervous systems

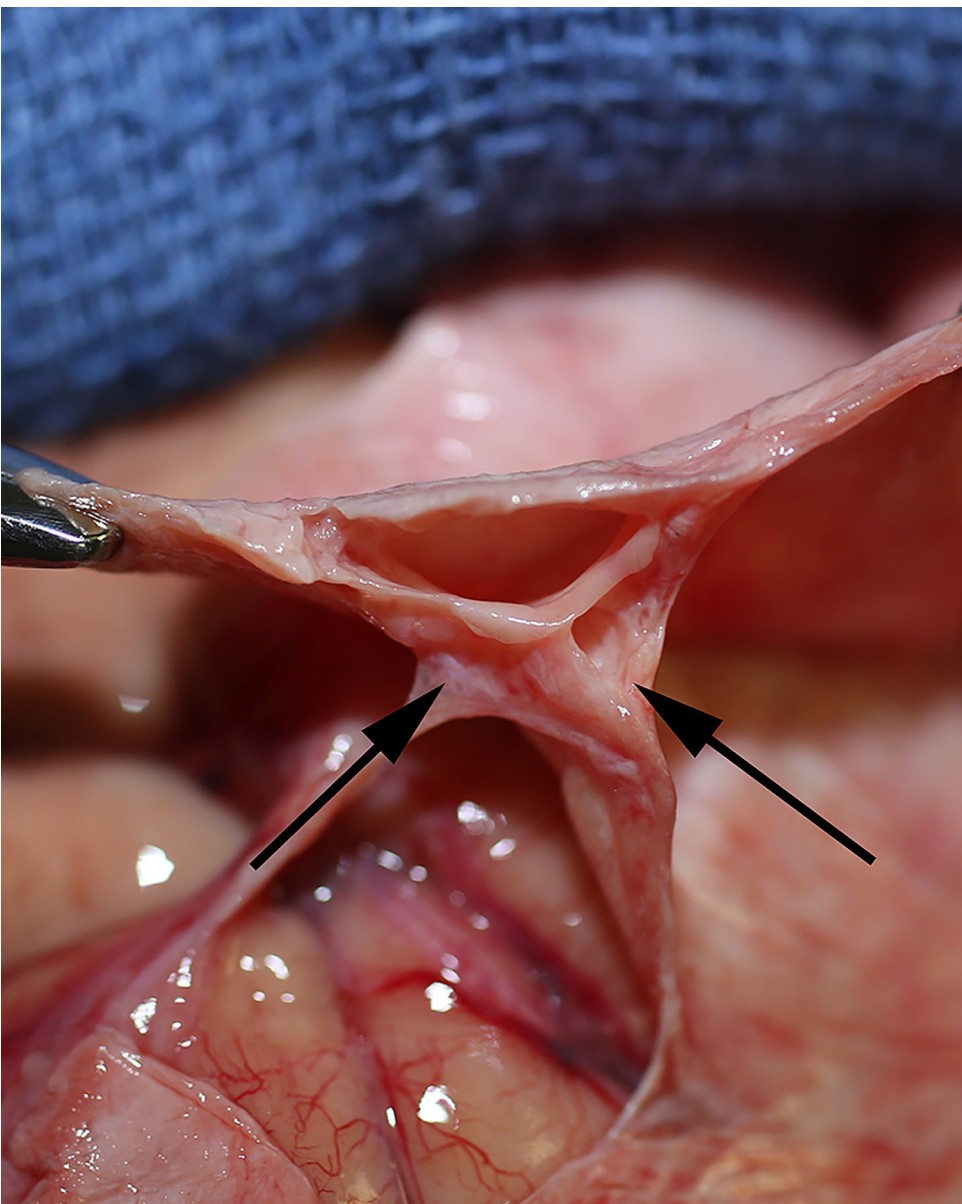

**Fig 7. CSF channels in the sagittal sinus.** Clamps suspend dura and show the empty venous sinus and bilateral CSF channels (arrows). The right side is widely patent, whereas the left has been obliterated by infiltrating carcinoma. The arachnoid meninges travel to these CSF channels.

(Figs 23 and 24). The primary path for CSF drainage is humans may be via the CSF canalicular system [15], with secondary drainage by extra-dural lymphatics to the scalp as described in mice [22].

The identification of the CSF canalicular system has implications for basic anatomy, surgery, and neuroscience. It redefines the anatomy of the sagittal sinus. The definition of what structures travel in the carotid sheath could be amended to include the terminal CSF drainage of the brain [23]. The observation of retrograde fill of the arachnoid meninges will require further study to identify potential CSF channels in that layer. Sappey's rule may need to be amended if further molecular analysis concludes that CSF channels are distinct from lymphatics and blood vessels [24].

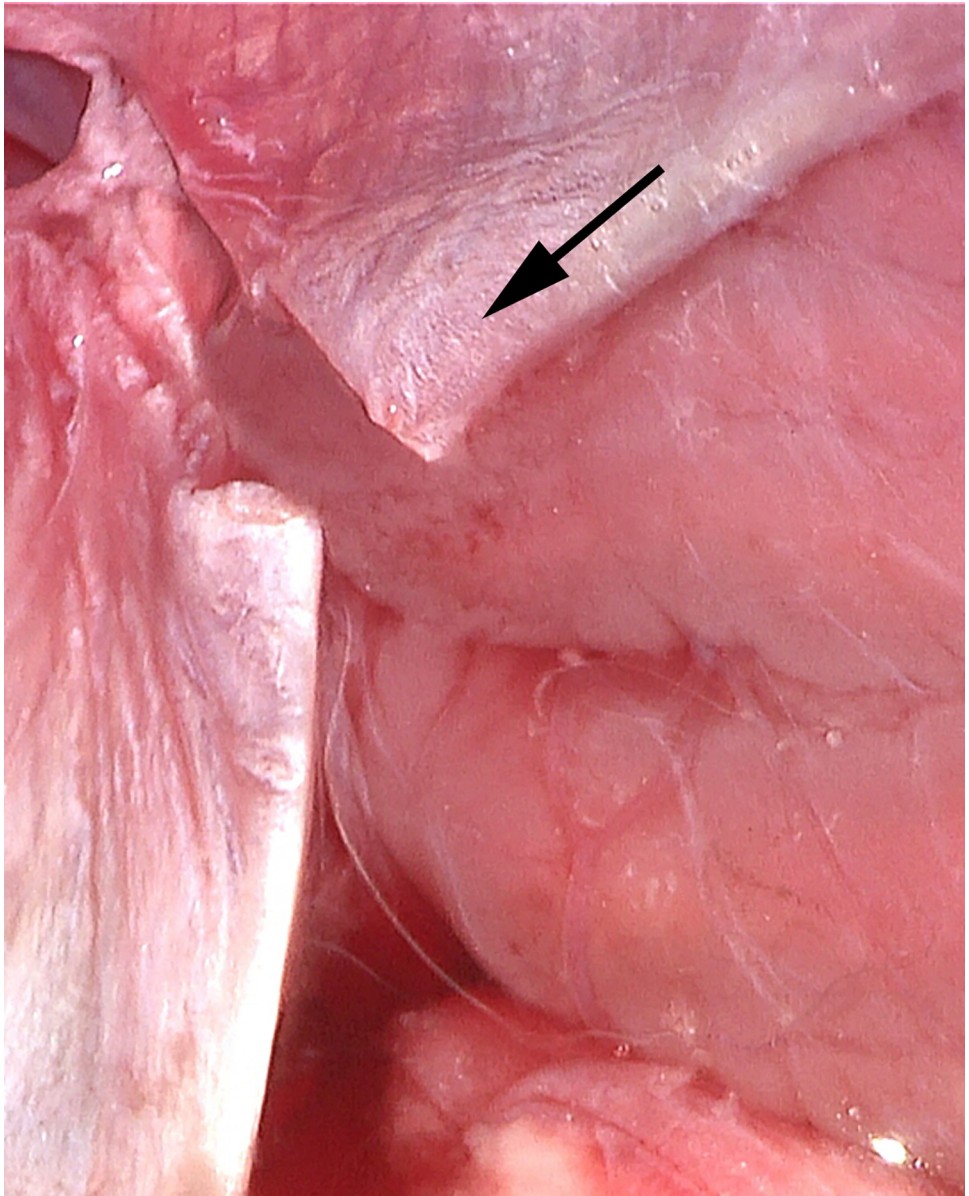

**Fig 8. Injection of CT contrast into CSF drainage.** The CSF drainage was cannulated and injected with CT contrast (injection was from superior to inferior, arrow) in this 60's year-old male specimen.

Evidence suggests that dysregulation of CSF drainage may be related to the etiology of neurocognitive disorders such as Alzheimer's disease [25, 26]. Certainly, dysregulation of flow in the CSF canalicular system may play a role in the etiology of hydrocephalus and possibly subarachnoid space enlargement noted in acceleration-deceleration traumatic brain injury [1, 2]. Astronauts subjected to prolonged microgravity may experience changes in vision, a disorder referred to as spaceflight-associated neuro-ocular syndrome (SANS). SANS may be related to dysregulation of CSF flow [27–29] and any possible role of the CSF canalicular system in SANS could be investigated.

CSF drainage is gravity dependent. Posture affects intracranial CSF volume and pressure, with *cerebral ventricular volume being lower in the upright position* (i.e. improved drainage)

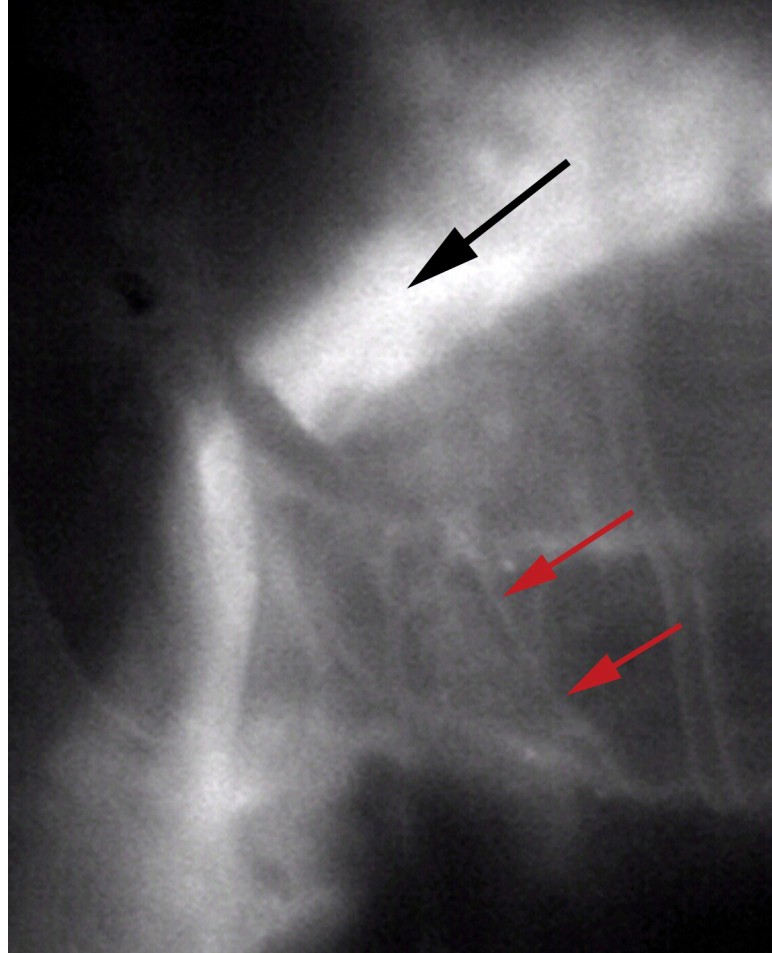

**Fig 9. Fluoroscopic imaging of CSF flow.** Fluoroscopy shows contrast traveling in CSF channels from the sagittal sinus to the cranial base. There appears to be some retrograde flow into CSF channels in the arachnoid meninges (red arrows).

[30]. The reverse has been noted after spaceflight. Cerebral ventricular volume may be increased in low-gravity environments or after return to 1 G [31]. Some authors propose the increased optic nerve sheath thickness is explained by stasis of flow in CSF channels found in the optic nerve dura (Fig 23) [32]. It is interesting that astronauts with SANS and Alzheimer's patients may both exhibit retinal vasculopathy and optic nerve disease [33–36]. Further research is needed.

## Future research

This work suggests potential avenues for future research. (1) The anatomic path of CSF flow along the cranial base will need to be mapped more accurately. (2) The molecular biology of the CSF canalicular system needs further work. (3) These findings suggest a reappraisal of the structure and concept of arachnoid villi. (4) This study has identified a novel anatomical structure that may be the primary path for CSF drainage of the human brain. The next question is as follows: *is there a disease process that affects the terminal CSF drainage that plays a role in the etiology of human disease*? Many other questions are suggested by this anatomical research.

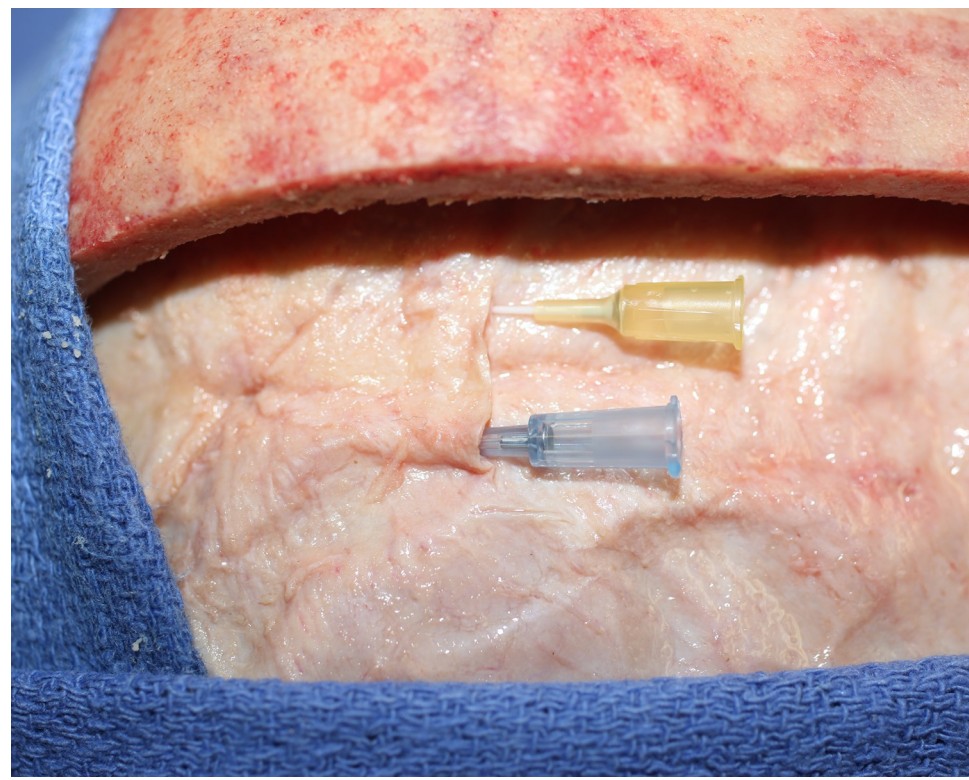

**Fig 10. Cannulation of CSF channels.** The CSF channels were cannulated in this 60's year old male specimen using a 22 gauge angiocatheter (blue) for the left system, and a 24 gauge angiocatheter (yellow) for the right system.

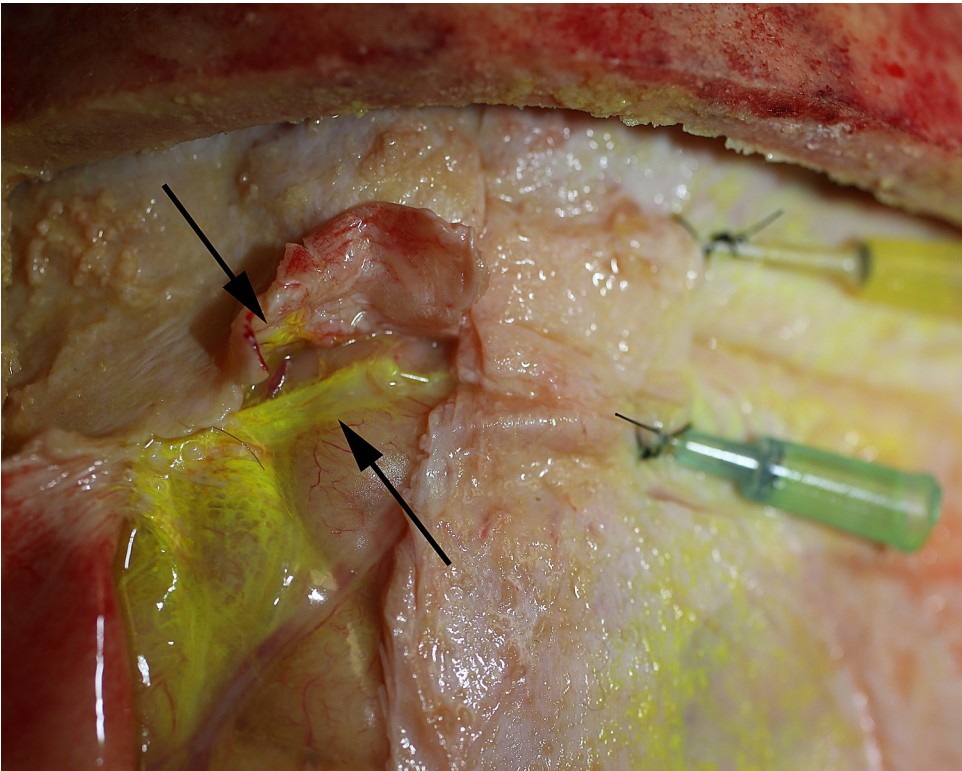

**Fig 11. Fluorescent dye injection.** Cospheric™ fluorescent polymer 1–5μ microparticles were injected into the CSF channels after which the sagittal sinus was transected. Flow was noted in both sides (arrows).

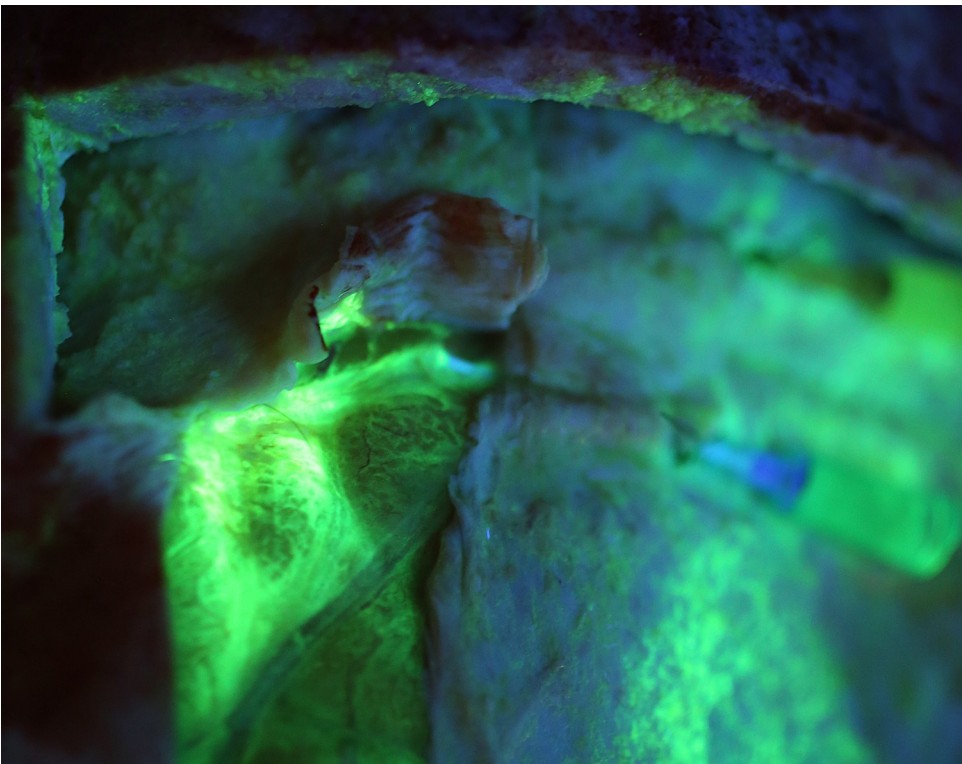

**Fig 12. Fluorescent imaging of flow in CSF channels.** UV fluorescence documents flow within these CSF channels. Note the backflow into the arachnoid meninges (green fluorescent blush of tissue beneath left CSF system).

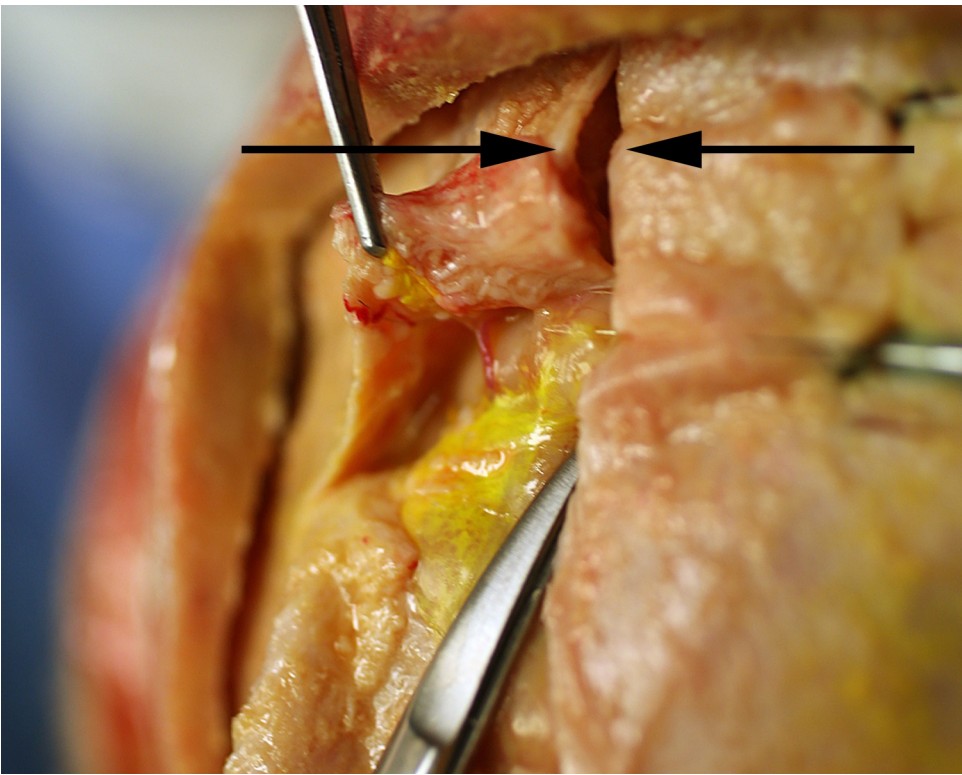

**Fig 13. CSF injection does not extravasate into veins.** When the CSF channels were injected, there was no filling of the venous system. The venous sagittal sinus (black arrows) did not fluoresce. Note side branches (red vessels) of the sagittal sinus did not fill with fluorescent dye.

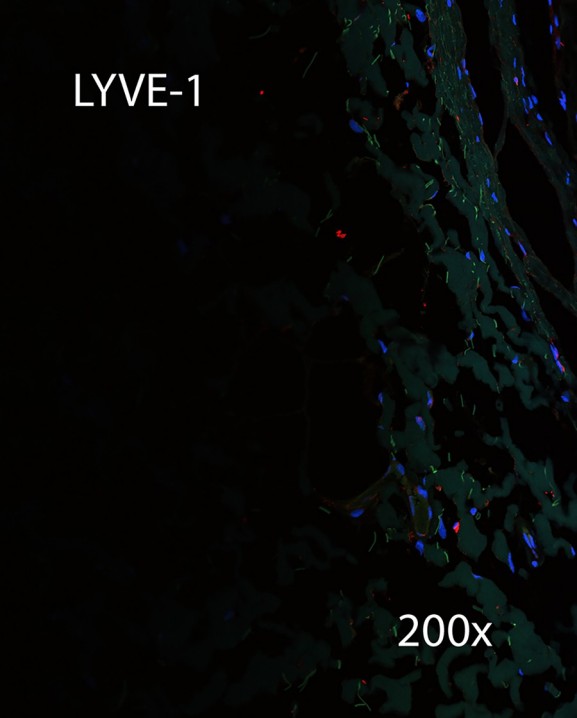

**Fig 14. IHC for LYVE-1.** IHC was negative for LYVE-1.

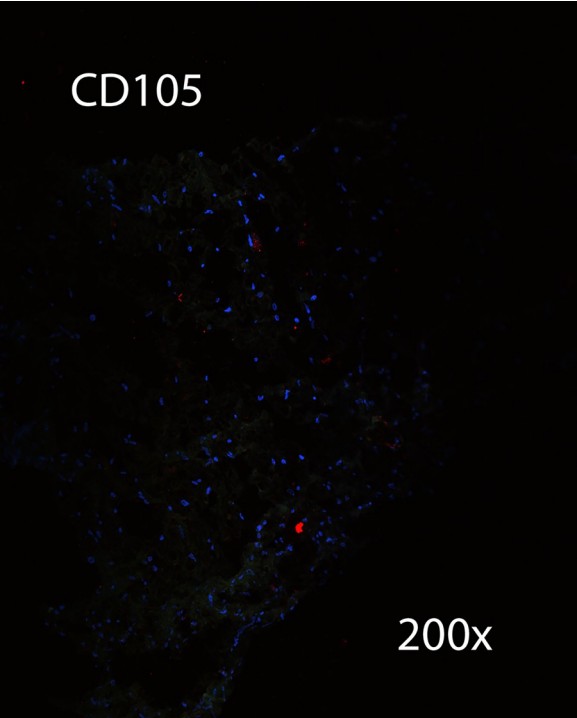

**Fig 15. IHC for CD105.** IHC was negative for vascular endothelial marker CD105.

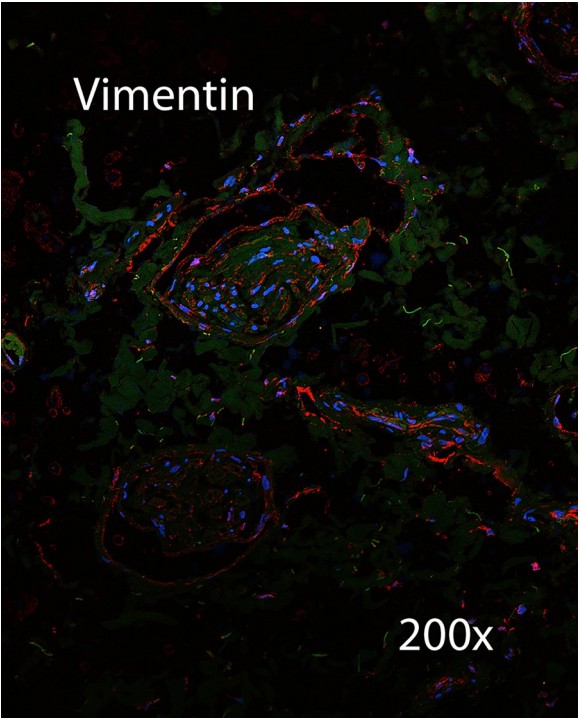

**Fig 16. IHC for vimentin.** IHC was positive for the type-3 neurofilament protein vimentin.

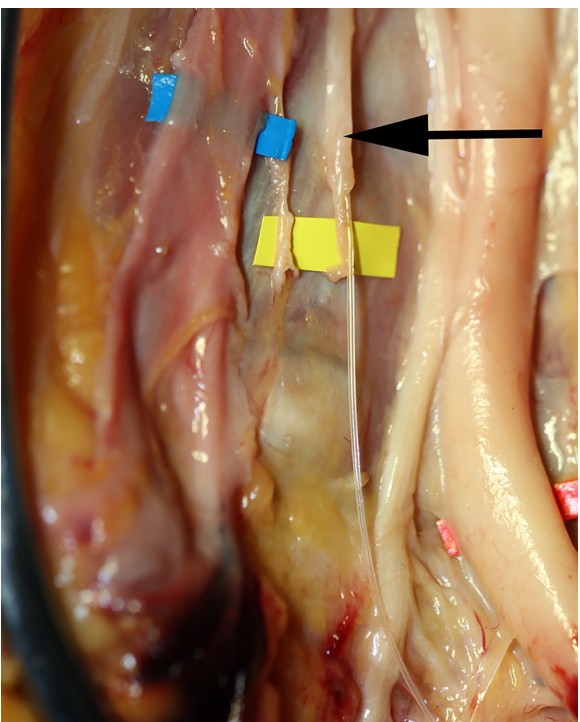

**Fig 17. Cervical CSF system.** Neck dissection in 60's year-old female specimen verifies the terminal CSF drainage in the neck (arrow). This travels within the adventia of the internal jugular vein (on blue marker). The carotid artery and vagus nerve are to the right of the CSF system.

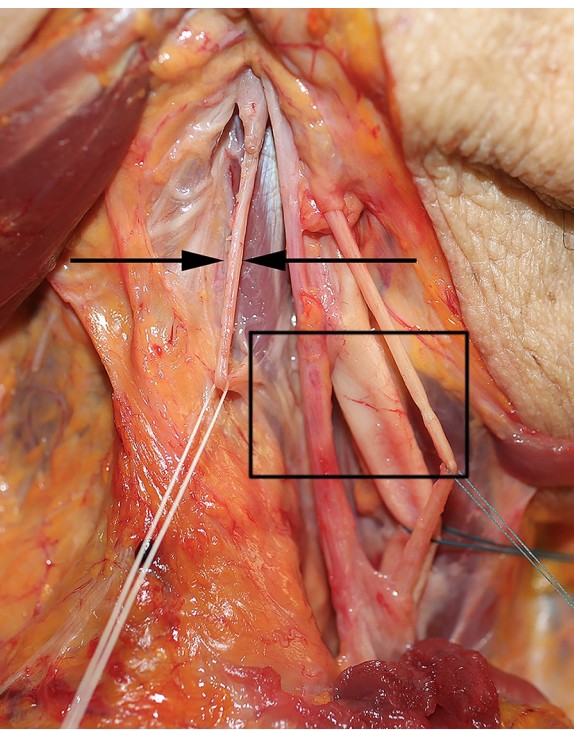

**Fig 18. Cervical CSF system.** Neck dissection in an 80's year-old female specimen verifies the terminal CSF drainage system in the neck (arrow). The structures of the carotid sheath are seen in the rectangle and include (from left to right) the internal jugular vein, carotid artery, and vagus nerve.

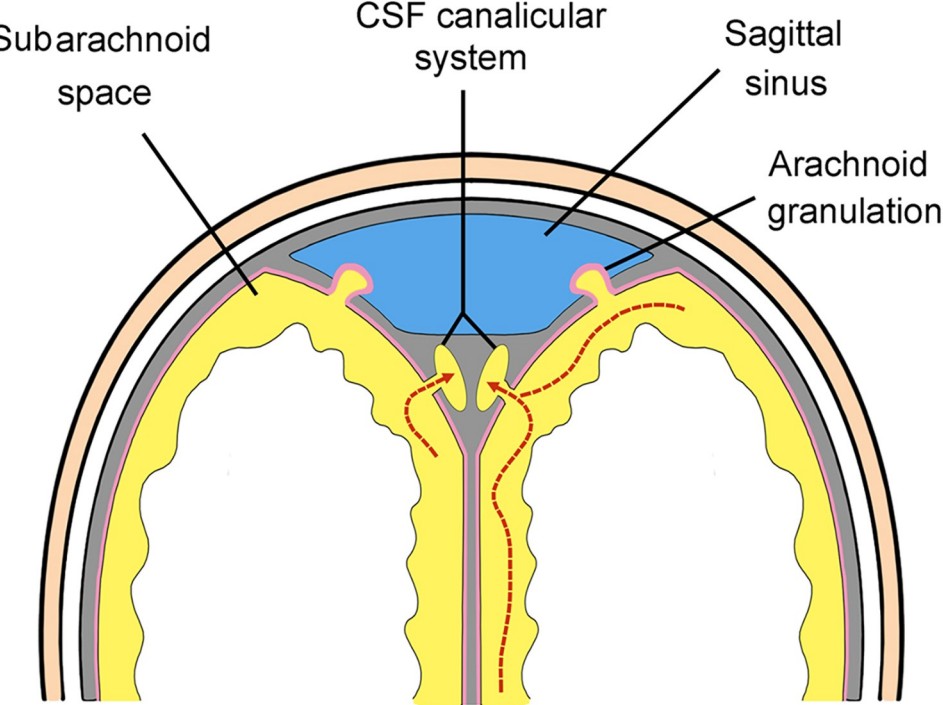

**Fig 19. A cross-section diagram of the CSF canalicular system.** The CSF canalicular system is located on either side of the venous sagittal sinus. CSF (yellow) flows from the subarachnoid space into CSF channels.

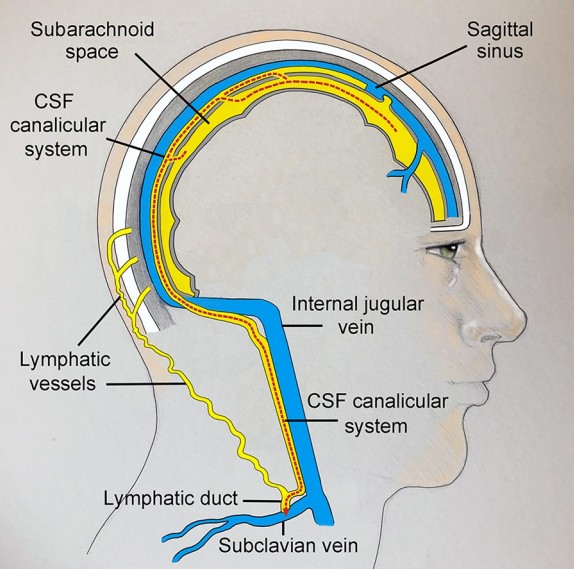

**Fig 20. Sagittal view of the CSF canalicular system.** The CSF canalicular system provides an anatomical route for CSF (yellow) to drain from the subarachnoid space directly to the subclavian vein.

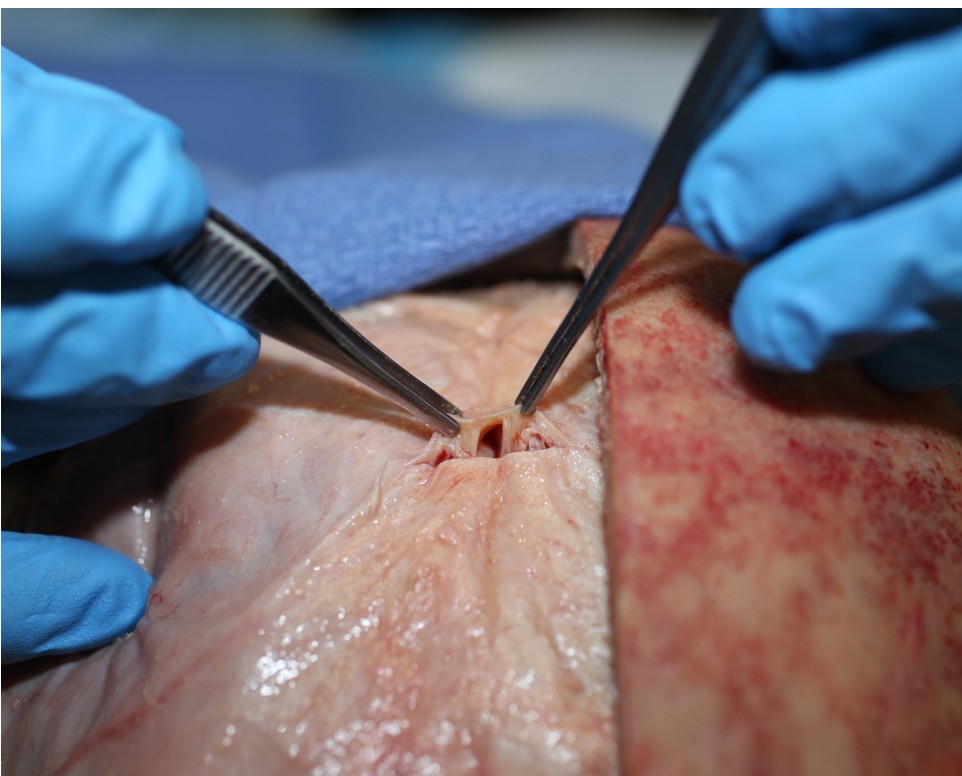

**Fig 21. The CSF canalicular system is a consistent feature of sagittal sinus anatomy.** It is straightforward to identify the CSF canalicular system that is located on either side of the venous sagittal sinus.

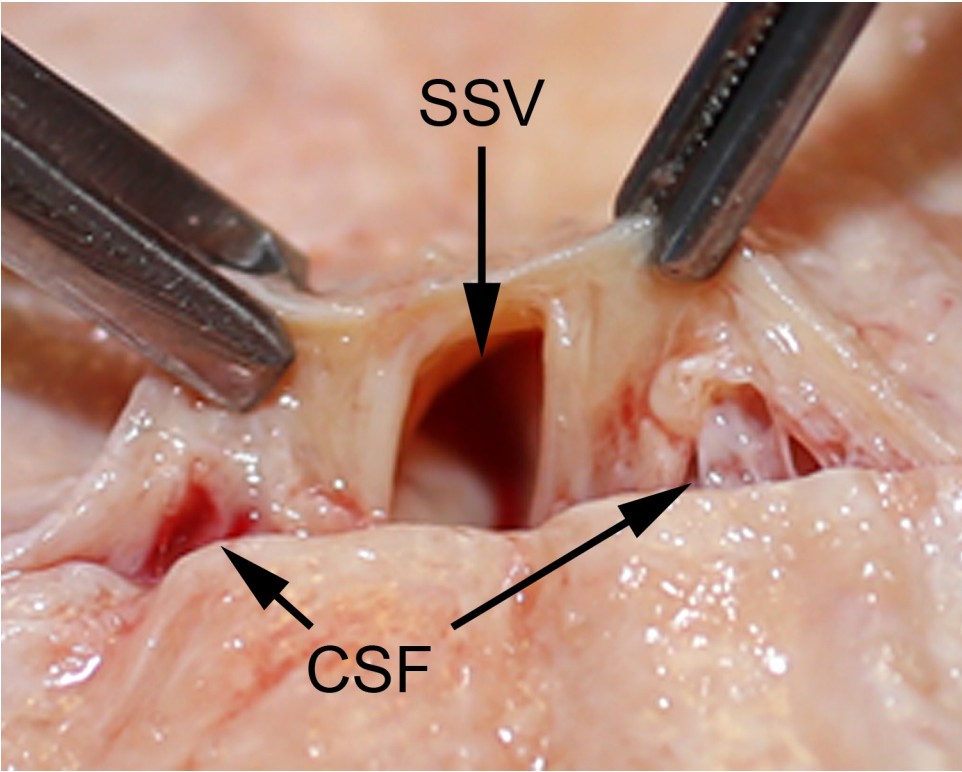

**Fig 22. The location of CSF canalicular system relative to the venous sagittal sinus.** Macro view shows the centrally-located sagittal sinus vein (SSV) accompanied by CSF channels (CSF) located on either side.

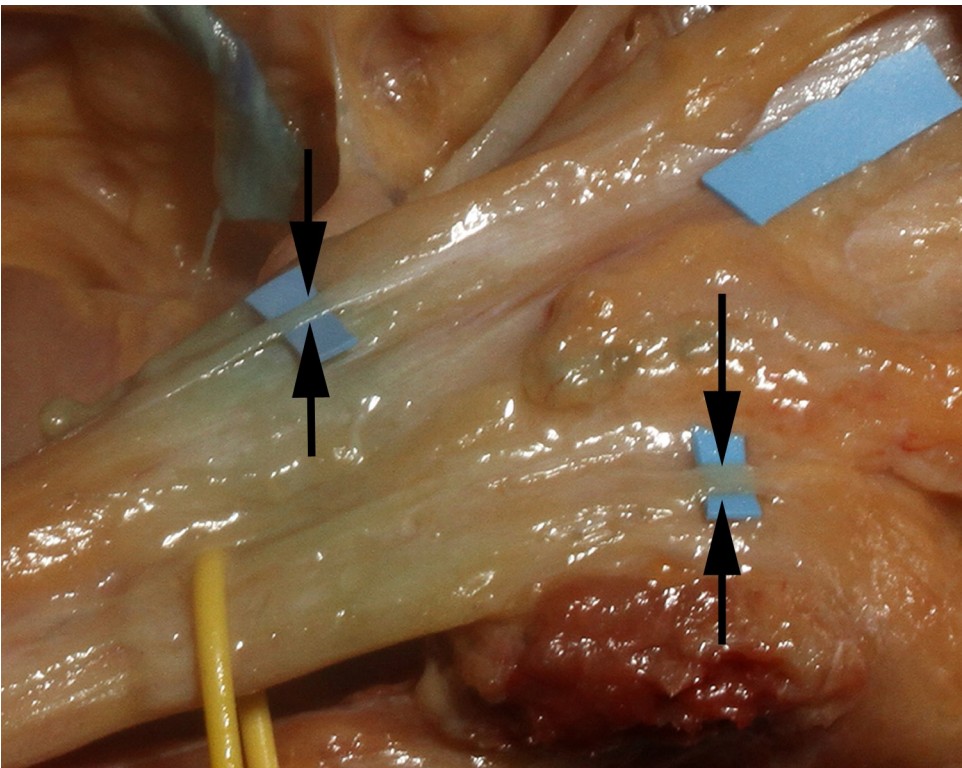

**Fig 23. The CSF canalicular system of nerves.** Draining CSF channels (arrows) on nerve travel within the outer layer of nerve (epineurium) that is embryologically analogous to the outer meninges of brain (dura). The terminal drainage of CSF in nerves is to the subclavian vein.

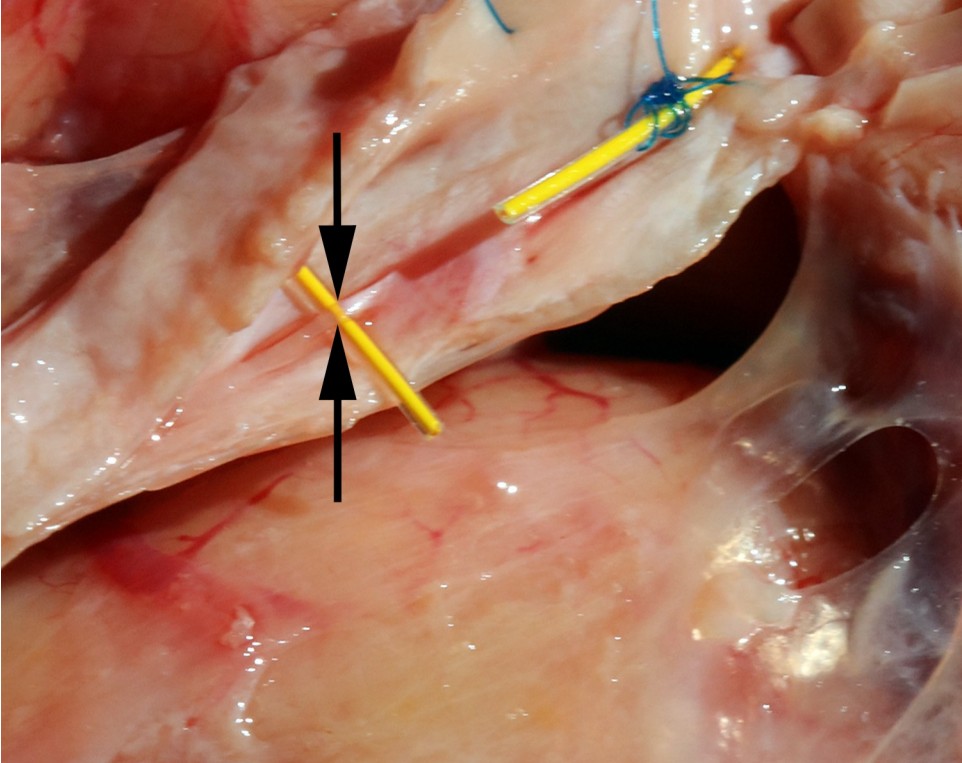

**Fig 24. Similarity of CSF channels in the central and peripheral nervous systems.** Note the similarity in appearance of CSF channels in the sagittal sinus (arrow) to those found in human nerve (Fig 23).

**Table 1. Characterization of the CSF canalicular system.** The CSF canalicular system provides an anatomic route for CSF drainage directly from the arachnoid meninges to the subclavian vein, and is independent of the venous sagittal sinus.

| | |
|---|---|
| Privileged | Does not involve intermediary lymphatics or blood vessels |
| Direct route | From arachnoid meninges to the subclavian vein |
| May be primary path | Secondary path to scalp lymphatics and lymph nodes |
| Drains to subclavian vein | CSF recycled into the vascular circulation |
| Analogous to nerves | Terminal CSF drainage of both brain and nerves is subclavian vein |
| Canalicular structure | Channels are embedded in surrounding tissue |
| Plexiform | Travel as a group of channels |
| Lack valves / muscular wall | Usually single-cell layer thickness |

## Conclusions

This paper identifies a novel anatomical path for CSF drainage of the human brain. The identification of the CSF canalicular system has implications for anatomy, surgery, and neuroscience, and highlights the continued importance of gross anatomy to medical research and discovery.

## Acknowledgments

The author thanks the donors of the UTSW Willed Body Program and their families. Without their generous gift this research would not be possible.

The author thanks the entire staff of the UTSW Willed Body Program for their help on this and all of our anatomical projects.

The author acknowledges Ronald M. Howorth MD's help in validating this model and performing the fluorescence injection and imaging.

## Author Contributions

**Conceptualization:** Joel E. Pessa.

**Formal analysis:** Joel E. Pessa.

**Investigation:** Joel E. Pessa.

**Methodology:** Joel E. Pessa.

**Writing – original draft:** Joel E. Pessa.

**Writing – review & editing:** Joel E. Pessa.

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
