## [Decision Letter · Decision Letter 0]

8 Feb 2023

PONE-D-23-00454A novel extra-venous pathway drains cerebrospinal fluid (CSF) from the subarachnoid space to the thoracic duct: the CSF canalicular system. Implications for spaceflight-associated neuro-ocular syndrome.PLOS ONE

Dear Dr. Pessa,

Thank you for submitting your manuscript to PLOS ONE. After careful consideration, we feel that it has merit but does not fully meet PLOS ONE’s publication criteria as it currently stands. Therefore, we invite you to submit a revised version of the manuscript that addresses the points raised during the review process. Please see comments below.

We look forward to receiving your revised manuscript.

Kind regards,

Alvan Ukachukwu, MD, MSc.GH

Academic Editor

PLOS ONE

Journal Requirements:

3. We noted in your submission details that a portion of your manuscript may have been presented or published elsewhere:

"Previous work was cited in the manuscript as a basis to complete this anatomic pathway."

5. Please amend your manuscript to include your abstract after the title page.

**Additional Editor Comments:**

The authors present a novel description of an alternative CSF pathway with significant implications for modifying our current understanding of CSF flow. The reviewers agree on the novelty of this interesting finding but have several suggestions to strengthen the manuscript. The major ones are:

- Addition of an anatomic diagram to clearly show this new description.

- Addition of a diagram showing the current understanding of CSF flow, and comparing with the new description.

- Description of lymphatic biomarkers used in this study to distinguish the CSF pathway from lymphatics (with visualization).

- Discussing the importance of this new pathway on common gravitational effects on standing, maintaining an erect posture, etc, as the connection to weightlessness and space flight may be tenuous.

Reviewers' comments:

Reviewer's Responses to Questions

**Comments to the Author**

1. Is the manuscript technically sound, and do the data support the conclusions?

Reviewer #1: Partly

Reviewer #2: Yes

2. Has the statistical analysis been performed appropriately and rigorously? 

Reviewer #1: N/A

Reviewer #2: N/A

3. Have the authors made all data underlying the findings in their manuscript fully available?

Reviewer #1: Yes

Reviewer #2: Yes

4. Is the manuscript presented in an intelligible fashion and written in standard English?

Reviewer #1: No

Reviewer #2: Yes

5. Review Comments to the Author

Reviewer #1: This is an interesting article proposing a new CSF drainage pathway. The presence of such a pathway has some interesting implications. The main issues with the article are:

1. It's very difficult to interpret was is going on from the images. The authors should include an anatomic drawing that guides the reader to what will be seen in the images. Perhaps a diagram could also be included that outlines the understand of CSF drainage before this finding and how that understanding should change based on this new info.

2. Before jumping to conclusions about weightlessness, the authors might want to think about what this means for postural changes. The canalicular system may be gravity dependent but so are the veins and arteries. What are the implications of this for the upright vs. supine posture?

Reviewer #2: Very interesting and well written report describing the vasculature draining the CSF.

1. Please specify which lymphatic biomarkers were assessed to verify that the CSF vessels were not lymphatic vessels. Were labeled lymphatic vessels observed in the same histologic slides containing the CSF vessels? Including such an image could provide more evidence that these are indeed not lymphatic vessels.

A couple of other very minor comments:

1. Figure 4: you have two types of arrows in figure (long and short) but only refer to arrows in the legend, please specify in the legend which arrows are being referenced.

2. While indocyanine green fluorescence imaging is technically an 'infrared' technique, I recommend updating the text to refer to it as a 'near-infrared' technique to avoid potential confusion with thermal imaging which is often referred to as infrared imaging.

6. PLOS authors have the option to publish the peer review history of their article (what does this mean?). If published, this will include your full peer review and any attached files.

Reviewer #1: No

Reviewer #2: No

---

## [Author Response · Author response to Decision Letter 0]

19 Mar 2023

Response to Reviewers,

Alvan Ukachukwu, MD, MSc.GH

Academic Editor

PLOS ONE

1. The paper was re-written according to the PLOS ONE guidelines and formatting.

2. This study was reviewed by the IRB committee and it was determined that it did not require oversight or approval. This is stated in Materials and Methods.

3. The identification of CSF channels in the sagittal sinus is the focus of this work and the main part of the study, and this has not been published in a peer-reviewed journal. In addition, we have not described the intracranial CSF system. The prior work on the CSF drainage of the neck was verified in this paper with new dissections, and only included to complete the circuit from sagittal sinus to subclavian vein. We re-wrote the paper to emphasize the new features and to avoid “dual publication”. 

4. We changed the title to simplify the message for a general audience. The identification of CSF channels in the sagittal sinus is a novel finding to the best of our knowledge We performed our historical review at Countway Medical Library Historical Books Collection. 

5. The manuscript was amended to include the abstract after the Title Page. 

Thank you for allowing us to re-submit a major revision. The paper was completely re-written to address the above comments and to hopefully make it interesting to the general scientific and medical community. 

Response to Additional Editor Comments:

1. Medical illustrations with coronal and sagittal views were included to show the new model of CSF drainage as Figs 9a and 9b. 

2. Medical illustrations with coronal and sagittal views were included in the INTRODUCTION to show the accepted model of CSF drainage as Figs 1a and 1b. 

3. We included our confocal imaging of LYVE-1 as the lymphatic marker, and CD105 for our vascular endothelial cell marker. Podoplanin was no different from LYVE-1, so was not included to be concise. The positive vimentin image was included. These are Figs 7a-c. We re-wrote the paper to focus on the gross anatomy, and avoided making definite claims that CSF channels are not lymphatics. We are aware of Loveau’s and others’ work on meningeal lymphatics and want to avoid this debate and focus on the identification of new anatomy in the sagittal sinus. 

4. We discussed the effects of posture on CSF drainage and included reference 30. However, if possible, we would like to include a brief discussion of spaceflight-associated neuro-ocular syndrome because of the similar effects of microgravity and the supine position on CSF volume (both increase cerebral ventricular CSF volume). We included references 27-29 and 31 to clarify this topic. 

Thank you again for allowing us to revise this manuscript. 

Reviewer #1:

1. It's very difficult to interpret what is going on from the images. The authors should include an anatomic drawing that guides the reader to what will be seen in the images. Perhaps a diagram could also be included that outlines the understand of CSF drainage before this finding and how that understanding should change based on this new info. RESPONSE: We included medical illustrations of the accepted model of CSF drainage (Figs 1a and 1b) and the amended model of CSF drainage (Figs 9a and 9b) based on our findings. 

2. Before jumping to conclusions about weightlessness, the authors might want to think about what this means for postural changes. The canalicular system may be gravity dependent but so are the veins and arteries. What are the implications of this for the upright vs. supine posture? 

RESPONSE: We included a discussion of how posture affects CSF volume with citation of reference 30. We also briefly discussed spaceflight-associated neuro-ocular syndrome because cerebral ventricular volume increases both in the supine position and in microgravity environments. We thought this is important because previous work (references 27-29) focused on sagittal sinus venous volume. Because CSF is privileged and independent of venous drainage, we thought this might be helpful to researchers in that field. 

Reviewer #2: Very interesting and well written report describing the vasculature draining the CSF.

1. Please specify which lymphatic biomarkers were assessed to verify that the CSF vessels were not lymphatic vessels. Were labeled lymphatic vessels observed in the same histologic slides containing the CSF vessels? Including such an image could provide more evidence that these are indeed not lymphatic vessels. 

 RESPONSE: Confocal laser imaging of LYVE-1 was included to show that it was negative for the biopsy of the cervical CSF canalicular system. Podoplanin (D2-40) was also negative but was not included for brevity. Negative CD105 was shown as our vascular endothelial marker. We included an image of a positive vimentin slide since this may be a consistent biomarker for CSF channels in the dura and in human peripheral nerve. The paper was re-written to emphasize the gross anatomy and to leave a more detailed molecular characterization for another project. 

2. Figure 4: you have two types of arrows in figure (long and short) but only refer to arrows in the legend, please specify in the legend which arrows are being referenced. RESPONSE: Fig 4 was re-labelled with only one arrow and now appears as Figs 3a and 3b. 

3. While indocyanine green fluorescence imaging is technically an 'infrared' technique, I recommend updating the text to refer to it as a 'near-infrared' technique to avoid potential confusion with thermal imaging which is often referred to as infrared imaging. RESPONSE: Thank you for pointing this out, we mislabeled the image. The near-infrared description was for another image (not used) and this was CT imaging after injection of iohexol (OmnipaqueTM) dye. The Figure Legends for Figs 5a and 5b were corrected. 

Thank you for reviewing our manuscript and for your criticisms and suggestions. 

Sincerely,

Joel Pessa MD and Ronald Hoxworth MD MBA

---

## [Decision Letter · Decision Letter 1]

11 Apr 2023

PONE-D-23-00454R1Identification of a novel path for cerebrospinal fluid (CSF) drainage of the human brainPLOS ONE

Dear Dr. Pessa,

Thank you for submitting your manuscript to PLOS ONE. After careful consideration, we feel that it has merit but does not fully meet PLOS ONE’s publication criteria as it currently stands. Therefore, we invite you to submit a revised version of the manuscript that addresses the points raised during the review process.

ACADEMIC EDITOR: Please see comments below.==============================

We look forward to receiving your revised manuscript.

Kind regards,

Alvan Ukachukwu, MD, MSc.GH

Academic Editor

PLOS ONE

Journal Requirements:

Additional Editor Comments:

The authors have responded satisfactorily to the initial reviewers' feedback. However, one of the reviewers has a few minor corrections which the authors need to address.

Reviewers' comments:

Reviewer's Responses to Questions

**Comments to the Author**

1. If the authors have adequately addressed your comments raised in a previous round of review and you feel that this manuscript is now acceptable for publication, you may indicate that here to bypass the “Comments to the Author” section, enter your conflict of interest statement in the “Confidential to Editor” section, and submit your "Accept" recommendation.

Reviewer #1: (No Response)

Reviewer #2: All comments have been addressed

2. Is the manuscript technically sound, and do the data support the conclusions?

Reviewer #1: Yes

Reviewer #2: Yes

3. Has the statistical analysis been performed appropriately and rigorously? 

Reviewer #1: N/A

Reviewer #2: N/A

4. Have the authors made all data underlying the findings in their manuscript fully available?

Reviewer #1: Yes

Reviewer #2: Yes

5. Is the manuscript presented in an intelligible fashion and written in standard English?

Reviewer #1: Yes

Reviewer #2: Yes

6. Review Comments to the Author

Reviewer #1: The authors have addressed the main comments very well. There are just two changes about the references to SANS. The data showing increased ventricular volume are from imaging done AFTER spaceflights. The changes in ventricular volume could have occurred either in space or during the re-adaptation to one-G. With that in mind change:

"SANS is thought to be related to dysregulation of CSF flow (27-29). Is SANS related to decreased

245 drainage in the CSF canalicular system?"

to

SANS may be related to dysregulation of CSF flow (27-29) and any possible role of the CSF canalicular system in SANS could be investigated.

and change:

"The reverse is noted during spaceflight, where cerebral ventricular volume is increased in low-gravity environments (31). The increased thickness of the optic nerve sheath is explained by stasis of flow in CSF channels found in the optic nerve dura (Fig 11a) (32)."

to

The reverse has been noted after spaceflight. Cerebral ventricular volume may be increased in low-gravity environments or after return to 1 G (31). Some authors propose the increased optic nerve sheath thickness is explained by stasis of flow in CSF channels found in the optic nerve dura (Fig 11a) (32).

Reviewer #2: (No Response)

7. PLOS authors have the option to publish the peer review history of their article (what does this mean?). If published, this will include your full peer review and any attached files.

Reviewer #1: No

Reviewer #2: No

---

## [Author Response · Author response to Decision Letter 1]

12 Apr 2023

Response to Reviewers

Additional Editor Comments

The authors have responded satisfactorily to the initial reviewers' feedback. However, one of the reviewers has a few minor corrections which the authors need to address.

Response: The two changes requested by Reviewer #1 have been made and inserted into lines 243-245 and into lines 249-252. The references have been verified. Thank you for reviewing our manuscript.

Reviewer #1

1. The data showing increased ventricular volume are from imaging done AFTER spaceflights. The changes in ventricular volume could have occurred either in space or during the re-adaptation to one-G. With that in mind change:

"SANS is thought to be related to dysregulation of CSF flow (27-29). Is SANS related to decreased drainage in the CSF canalicular system?"

to

”SANS may be related to dysregulation of CSF flow (27-29) and any possible role of the CSF canalicular system in SANS could be investigated.”

Response: Lines 243-245 have been changed to “SANS may be related to dysregulation of CSF flow (27-29) and any possible role of the CSF canalicular system in SANS could be investigated.” 

2. …and change: "The reverse is noted during spaceflight, where cerebral ventricular volume is increased in low-gravity environments (31). The increased thickness of the optic nerve sheath is explained by stasis of flow in CSF channels found in the optic nerve dura (Fig 11a) (32)."

to

”The reverse has been noted after spaceflight. Cerebral ventricular volume may be increased in low-gravity environments or after return to 1 G (31). Some authors propose the increased optic nerve sheath thickness is explained by stasis of flow in CSF channels found in the optic nerve dura (Fig 11a) (32).”

Response: Lines 249-252 have been changed to “The reverse has been noted after spaceflight. Cerebral ventricular volume may be increased in low-gravity environments or after return to 1 G (31). Some authors propose the increased optic nerve sheath thickness is explained by stasis of flow in CSF channels found in the optic nerve dura (Fig 11a) (32).”

Thank you for reviewing our manuscript and pointing out that changes could occur during re-adaptation to 1G. 

Reviewer #2: (No Response)

Response: Thank you for reviewing our manuscript.

---

## [Editor Report · Decision Letter 2]

19 Apr 2023

Identification of a novel path for cerebrospinal fluid (CSF) drainage of the human brain

PONE-D-23-00454R2

Dear Dr. Pessa,

We’re pleased to inform you that your manuscript has been judged scientifically suitable for publication and will be formally accepted for publication once it meets all outstanding technical requirements.

Kind regards,

Alvan Ukachukwu, MD, MSc.GH

Academic Editor

PLOS ONE
---

## [Editor Report · Acceptance letter]

26 Apr 2023

PONE-D-23-00454R2 

Identification of a novel path for cerebrospinal fluid (CSF) drainage of the human brain 

Dear Dr. Pessa:

I'm pleased to inform you that your manuscript has been deemed suitable for publication in PLOS ONE. Congratulations! Your manuscript is now with our production department. 

Kind regards, 

on behalf of

Dr. Alvan Ukachukwu 

Academic Editor

PLOS ONE